# BUT IS IT VALID? ENFORCING STRUCTURAL CONSTRAINTS ON GRAPH GENERATIVE MODELS

## ABSTRACT

The problem of graph generation using deep learning has received substantial attention in the recent years. When using graph generative models, one often faces the issue that the generated graphs do not respect hard constraints of the empirical distribution. A common challenge is to guarantee even basic structural properties of the generated graphs, such as connectedness or planarity. In this work, we propose ValiGraph, a graph generative method based on denoising diffusion, which guarantees the generation of graphs respecting a large family of structural properties. In addition we quantify the ability of the models in capturing topological information, we propose the use of extended persistent homology in the evaluation procedure. We show that ValiGraph is superior in capturing the distribution of graph structural features on several datasets.

## 1 INTRODUCTION

We propose **ValiGraph**, a one-shot graph generative model which is guaranteed to generate valid graphs that respect the hard structural constraints of the empirical distribution as long as these constraints can be expressed in terms of edge-addition and edge-deletion invariant properties. We show that enforcing such constraints greatly improves the model's ability to preserve graph topological properties of the distribution in question.

Respecting hards structural constraints is relevant both in theory and practice: In theory, preserving the support of the distribution is an attractive basic property for a generative model. In practice, graph validity often depends on preserving graph topology: Graph skeletons extracted from 2d biological images (Schaadt et al., 2020) or public transport networks (Háznagy et al., 2015) have to be planar, and social networks might show constrains on the nodes' degree. Examples are personal social networks which are often studied as ego-centered network (Hogan et al., 2023; Brea Perry & Small, 2023) or patients and healthcare workers interaction, whose degree might be limited to avoid too many contacts (Jang et al., 2019; Adhikari et al., 2019).

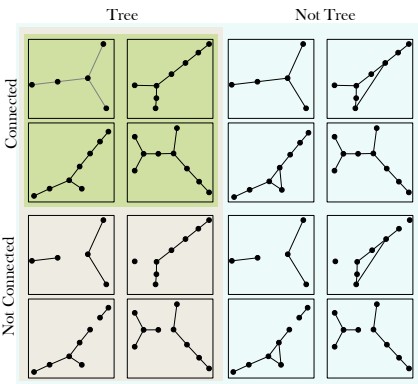

Figure 1: Examples of graphs with different structural properties invariant to edge-deletion (tree), and invariant to edge-addition (being connected). As opposed to previous methods ValiGraphis able to enforce both of these properties in the generated graphs.

Roughly speaking graph generative models can be divided into autoregressive and one-shot graph generative models, where one shot models generate all the graph's edges at once. As autoregressive models can check topological properties – e.g. being connected, planar, containing cycles – at each iterative sampling step, they are more capable at generating sampled graphs that satisfy the topological support of the empirical distribution. One-shot graph generative models typically do not have these properties by design, but comes with other advantages such as no need to impose an explicit node ordering, and it remains an open problem to ensure that the generated graphs are valid and respect the properties of the true graph distribution.

Discrete denoising diffusion has proved very effective as State-of-the-art one-shot graph generative models (Vignac et al., 2022; Madeira et al., 2024). These models typically augments each diffusion denoising step with topological information during training to increase the expressive power of the models, but this does not guarantee that topological properties of the graph distribution are maintained. Even a basic topological property such as connectedness is not guaranteed when sampling new graphs (see the illustration in Figure 1). This raises the question of how an inductive bias enforcing the topological constraints can be imposed on a one-shot graph generative model.

To validate generative models, two sets of metrics are commonly used: the validity, uniqueness and novelty (VUN) metrics on the generated dataset, and the metrics recently proposed by You et al. (2018) between the empirical and generated distributions. As Vignac et al. (2022), we evaluate the maximum mean discrepancy (MMD) between the distributions of edge degree, cluster coefficient, and the number of orbits with 4 nodes. Existing models can, however, perform well on these metrics even if the generated graphs contain a high percentage of graphs that do not respect the hard constraints of the empirical distribution. To bridge this gap it is thus crucial to identify additional topological features of graphs for the evaluation of graph generative models. **Our contribution:**

- **Documenting** that existing one-shot graph generative models struggle to capture topological properties of graph distributions, along with an illustration of the downstream issues that arise from this shortcoming.
- **Proposing** ValiGraph, building on a noise model which preserves graph structural properties that are invariant under edge-addition and edge-deletion. This enables us to enforce hard structural constraints on the generated graphs, thereby improving the model's ability to preserve the topological properties of graph distributions.
- **Proposing evaluation metrics** based on extended persistent homology that capture the extent to which the topological properties of graph distributions are preserved.

## 2 BACKGROUND

Generative models based on denoising diffusion are extremely effective for image generation, but can be used more widely: Discrete diffusion models are treated comprehensibly by Austin et al. (2021), whose modeling framework has been used in several state-of-the-art graph generative models (Jo et al., 2022; Vignac et al., 2022; Madeira et al., 2024).

**Discrete Denoising Diffusion:** We formulate the noise process using a Markov chain defined in a discrete state space. When modeling labeled graphs, the state space describe the nodes and edges labels. Let $\mathbf{z} = [z_1, ..., z_K]$, where $z_k \in \{0, 1\}$ and $\sum_{k=1}^{K} z_k = 1$ be the one-hot encoding of label $k$ out of total $K$ labels. If $z_k = 1$, we say that $\mathbf{z}$ is in state $k$. The forward diffusion process can now be formulated by defining transition matrices $\mathbf{Q}^1, \mathbf{Q}^2, ..., \mathbf{Q}^T$ for each time-step $t \in 1, ..., T$, where $[\mathbf{Q}^t]_{ij}$ is the probability of transitioning from state $i$ to state $j$. Thus, the forward process is:

$$q(\mathbf{z}^t|\mathbf{z}^{t-1}) = \mathbf{z}^{t-1}\mathbf{Q}^t \tag{1}$$

$$q(\mathbf{z}^t|\mathbf{z}) = \mathbf{z}\bar{\mathbf{Q}}^t, \tag{2}$$

where $\bar{\mathbf{Q}}^t = \mathbf{Q}^1\mathbf{Q}^2, ..., \mathbf{Q}^t$. Following standard Markov process terminology, we call a state $k$ *transient* if $\sum_{t=1}^{\infty} P(z_k^t = 1|z_k^0 = 1) < \infty$, that is, there is a non-zero probability of never returning to the state. The state is called *recurrent* if $\sum_{t=1}^{T} P(z_k = 1|z_k^0 = 1) \to \infty$ when $T \to \infty$, that is, $\mathbf{z}^t$ has state $k$ occur infinitely often. Lastly, an *absorbing* state refers to a state where $P(z_k^{t+1} = 1|z_k^t = 1) = 1$. While technically an absorbing state is also a recurrent state, we only use the term to refer to states that are recurrent, but not absorbing unless otherwise mentioned.

Using absorbing states to enforce structural information with a discrete model has been used for text (Devlin et al., 2018) and image generation (Austin et al., 2021) as well as graph generation. In the latter works, each state is chosen to correspond to a specific discrete label: the node- and edge-type. Picking one of these types to correspond to an absorbing state (e.g. the no-edge-state as in ConStruct (Madeira et al., 2024)), has the consequence that a trivial point-mass prior must be picked. This is a considerable limitation of the model. Here, we demonstrate how carefully designed transition matrices enable us to consider non-trivial prior distributions.

**Graph Representation:** We consider a graph with cat-
egorical attributes on nodes and edges and create one-hot
encoding $\mathbf{x}_i = [x_1, ..., x_{d_x}]$ and $\hat{\mathbf{e}}_{i,j} = [e_1, ..., e_{d_e}]$ of the $d_x$
node types and $d_e$ edge types respectively. As we use dense
matrices, the absence of an edge in a graph is always desig-
nated with its own type. For ease of notation, we assume that,
given an edge $\hat{\mathbf{e}}$, we have $e_1 = 1$ if and only if the edge is a
non-edge (i.e., absent). In addition to the $d_e$ edge states in-
duced by the encoding of the edge-type we add an additional
$K$ auxiliary states which serve as either absorbing or recur-
rent states. To formalize the assignment of states to edges,
we define a map $h\colon \hat{\mathbf{e}} \mapsto \mathbf{e} = [e_1, ..., e_{d_e}, e_{d_e+1}, ..., e_{d_e+K}]$,

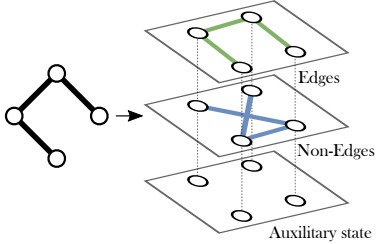

Figure 2

where $e_{d_e+k} = 0$ for $k \in \{1, ..., K\}$. These nodes and edges representations are then organized
in tensors $\mathbf{X} \in \{0,1\}^{n \times d_x}$ and $\mathbf{E} \in \{0,1\}^{n \times n \times (d_e+K)}$, where $\mathbf{X}[i] = \mathbf{x}_i$ and $\mathbf{E}[i,j] = \mathbf{e}_{i,j}$ for
$i, j \in \{1, ..., n\}$ with $n$ number of nodes. A graph $G \in \mathcal{G}$ is finally represented by the pair of
tensors $G = (\mathbf{X}, \mathbf{E}) \in \mathcal{G}$, where $\mathcal{G}$ denotes the space of all graphs. An illustration of the described
mapping of the edges in the case of a binary graph can be found in Figure 2. We can also con-
struct mappings in the opposite direction as follows. That is given an edge $\mathbf{e}$, we let $\pi_+$ define
the map that sends an edge with one-hot encoded attributes to the corresponding binary edge, i.e.
$\pi_+\colon \mathbf{e} \mapsto [e_1 + \sum_{i=d_e+1}^{d_e+K} e_i, \sum_{i=2}^{d_e} e_i]$, and $\pi_-\colon \mathbf{e} \mapsto [e_1, \sum_{i=2}^{d_e+K} e_i]$.

## 3 METHOD

We propose **ValiGraph**, a generalization of Madeira et al. (2024), to a graph generative model which
guarantees that generated graphs have specific properties invariant to edge-deletion and edge-addition.
Mathematically we formalize the notion of a property as an indicator function $\rho\colon \mathcal{G} \to \{0,1\}$ on the
space of graphs.

**Definition 3.1.** Let $\rho\colon \mathcal{G} \to \{0,1\}$ be a property. If for all graphs $G \in \mathcal{G}$ we have that $\rho(G') = \rho(G)$
for all $G' \in \mathcal{G}$ where $G' \subseteq G \implies \rho(G')$ then the property is invariant under *edge deletion*. Vice
versa if $\rho(G) = \rho(G')$ for all $G' \in \mathcal{G}$ where $G \subseteq G'$ the property is invariant under *edge addition*.

Given properties $\rho_1, ..., \rho_s$, we can also compose a new property $\rho = \prod_{1=i}^{s} \rho_i$. Abstract absorbing or
recurrent states used in all the noise models can be interpreted as applying an edge-deletion and an
edge-addition process in parallel. Letting $G \subseteq G'$ for $G, G' \in \mathcal{G}$ denote that $G$ is a subgraph of $G'$
we consider a sequence of graphs $G^0, G^1, ..., G^T$ sampled using a noise process outlined in Section
2. This induces an edge-deletion process $\pi_+(G^0) \supseteq \pi_+(G^1) \supseteq ... \supseteq \pi_+(G^T)$ because edges have
zero probability of returning from an auxiliary state to an edge state. Vice versa, we recover an
edge-addition process by noting that $\pi_-(G^0) \subseteq \pi_-(G^1) \subseteq ... \subseteq \pi_-(G^T)$. These two sequences of
sampled graphs contain information about the edge-deletion invariant, and edge addition invariant
properties respectively. Designing the noise model in this way has the remarkable consequence, that,
during inference, we can guarantee any graph property which can be composed by both edge-deletion
and edge-addition invariant properties. In Figure 3, we illustrate this process. Here, green edges
indicate the edges that are present in the graph and the blue edges indicate the edges of the graph
complement, i.e. edges which are not present in the original graph. At each timestep, there is a
chance of either edges- or non-edges transitioning to an absorbing state indicated by a gray edge. We
see, that eventually all edges have transitioned to the absorbing state. Likewise we see, that we can
recover an edge-deletion process by considering the green edges at each step of the noise process,
and an edge-addition process, by considering the graph complement of all blue edges at each step.

**Forward process:** As is common for discrete graph noise models, we apply noise independently to
each node and edge, which for a graph $G = (\mathbf{X}, \mathbf{E})$ reduces to sampling from the distributions:

$$q(G^t|G^{t-1}) = \prod_i q(\mathbf{x}_i^t|\mathbf{x}_i^{t-1}) \cdot \prod_{ij} q(\mathbf{e}_{ij}^t|\mathbf{e}_{ij}^{t-1}) \tag{3}$$

$$q(G^t|G) = \prod_i q(\mathbf{x}_i^t|\mathbf{x}_i) \cdot \prod_{ij} q(\mathbf{e}_{ij}^t|\mathbf{e}_{ij}), \tag{4}$$

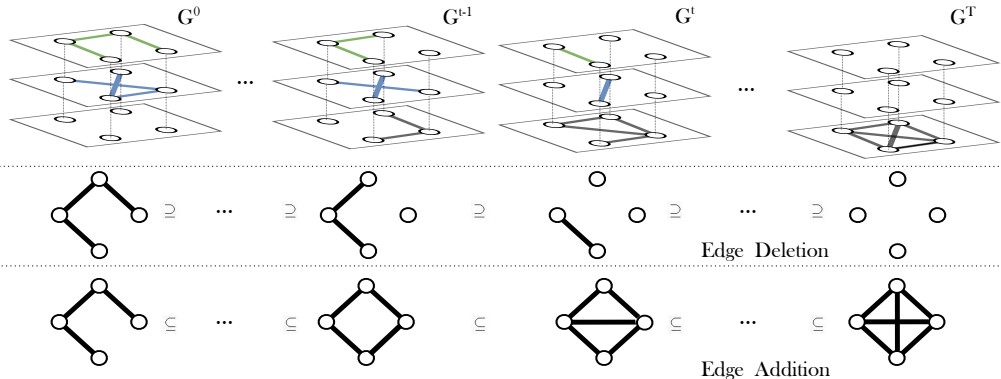

Figure 3: Illustration of what a noise process with a single absorbing state could look like when applied to a graph with four nodes, and three edges (green), and three non-edges (blue). The first row shows the noise process. The second and third rows show the corresponding edge-deletion and edge-addition processes.

where the node-wise noise process is defined by $q(\mathbf{x}_i^t|\mathbf{x}_i^{t-1}) = \mathbf{x}_i^{t-1}\mathbf{Q}_x^t$ and $q(\mathbf{x}_i^t|\mathbf{x}_i) = \mathbf{x}_i\bar{\mathbf{Q}}_x^t$ for transition matrices $\mathbf{Q}_x^t$ and $\bar{\mathbf{Q}}_x^t = \mathbf{Q}_x^1...\mathbf{Q}_x^t$. Analogously to the node-wise process, the edge-wise process is given by $q(\mathbf{e}_{ij}^t|\mathbf{e}_{ij}^{t-1})$ and $q(\mathbf{e}_{ij}^t|\mathbf{e}_{ij})$.

As we focus on creating a generative model which respects structural properties, we use the noise model of Vignac et al. (2022) on the nodes. The sequence of node-transition matrices is then:

$$\mathbf{Q}_x^t = \alpha_x^t\mathbf{I} + (1 - \alpha_x^t)\mathbf{1}_{d_x}\mathbf{p}_x, \tag{5}$$

where $\mathbf{I}$ is a $d_x \times d_x$ identity matrix and $\mathbf{p}_x[i]$ refers to the marginal probability of type $i$ in the training set. Additionally, $\alpha_x^t \in [0,1]$ is a noise schedule indexed over $t \in \mathbb{N}_0$, with $\alpha_x^0 = 1$ satisfying $\alpha_x^t \to 0$ when $t \to \infty$. In this work, the cosine noise schedule is used for node attributes. The advantage of formulating the transition matrix in this form, is that we can compute $\bar{\mathbf{Q}}_x^t = \bar{\alpha}_x^t\mathbf{I} + (1 - \bar{\alpha}_x^t)\mathbf{P}_x$, where $\bar{\alpha}_x^t = \prod_{i=0}^t \alpha_x^i$ whenever $\mathbf{P}_x^2 = \mathbf{P}_x$ and $\mathbf{P}_x$ is a stochastic matrix. We seek to write the edge transition matrices on this form. For the convenience of the reader, a proof is found in Appendix A.

**Defining suitable transition matrices:** In ConStruct, the noise process is an absorbing Markov chain, where the no-edge state is considered to be absorbing. In practice, this formulation yields an edge-deletion process. Some graph properties, e.g., planarity, do not change under edge deletion, and thus, this noise process has the advantage of preserving such properties of the input graphs. This admits a trivial limit distribution: the distribution of graphs with no edges.

We generalize this approach by formulating a noise model that preserves properties invariant to both edge-deletion and edge-addition, thus enforcing a common edge-addition invariant property like connectedness, while also preserving edge-deletion invariant properties (e.g., maximum possible node degree or planarity). Instead of considering an absorbing state Markov process, we add a number of additional place-holder states, which serve as either absorbing or recurrent states. As we investigate graph topology, we are mainly interested in binary graphs, but the approach equally applies to graphs with edge-attributes. The noise process for edge-attributes is defined on a form similar to $\mathbf{Q}_x^t$ as:

$$\mathbf{Q}_e^t = \alpha_e^t\mathbf{I} + (1 - \alpha_e^t)\mathbf{P}_e, \tag{6}$$

where $\mathbf{I}$ is a $(d_e + K) \times (d_e + K)$ identity matrix, and $\alpha_e^t$ denotes the noise schedule. We define $\alpha_e^t$ as the mutual information noise schedule(Austin et al., 2021; Sohl-Dickstein et al., 2015) (see Appendix B). Finally, we define:

$$\mathbf{P}_e = \begin{bmatrix} \mathbf{0} & \mathbf{T} \\ \mathbf{0} & \mathbf{S} \end{bmatrix}, \tag{7}$$

where $\mathbf{S}$ is a $K \times K$ stochastic matrix with absorbing (recurrent) states, and where $\mathbf{T}$ is a $d_e \times K$ stochastic matrix defined such that the entry $i, j$ of $(1 - \alpha_e^t)\mathbf{T}$ denotes the probability of transitioning to the absorbing (recurrent) state $j$ from state $i$. This formulation is advantageous as it offers great flexibility in the choice of $\mathbf{S}$, thus admitting non-trivial limit distributions and more complex priors, as opposed to the conventional approaches using only a single absorbing state. Below, we account for the noise models considered in this article and validate that they indeed have the desired properties.

**Single-state Absorbing Noise Model:** Letting $\mathbf{S}$ be a $1 \times 1$ identity matrix, and $\mathbf{T}$ be a $d_e \times 1$ matrix of ones results in a noise model with a single absorbing state. This noise model extends the absorbing noise model used in ConStruct to edge-addition invariant properties. The fact that $\mathbf{P}$ is idempotent is trivial in this case and consequently $\bar{\mathbf{Q}}^t$ can be computed in closed form as outlined above. We also see that $\mathbf{e}\bar{\mathbf{Q}}^t \to \mathbf{e}\mathbf{P}_e$ when $t \to \infty$. However, for all $\mathbf{e}$ we see that $\mathbf{e}\mathbf{P}$ is just the one-hot encoding of the absorbing state, and as such the limit distribution does not depend on the input. We refer to this model as SAS-ValiGraph.

**Multi-state Absorbing Noise Model:** Letting $\mathbf{S}$ be a $d_e \times d_e$ identity matrix and $\mathbf{T} = \mathbf{1}\mathbf{p}_e$ results in an absorbing state model with $d_e$ absorbing states. Here, we have let the transition probability to each recurrent state to be proportional to the empirical marginal probability of the specific edge type. Again, it follows that $\mathbf{P}$ is idempotent, by exploiting the fact that the identity matrix is idempotent and that $\mathbf{T}^2 = \mathbf{1}\mathbf{p}_e\mathbf{1}\mathbf{p}_e = \mathbf{T}$, since $\mathbf{p}_e\mathbf{1} = 1$ by construction. Here, the limit distribution is $\mathbf{e}\mathbf{P} = \mathbf{p}_e$ for all one-hot-encoded edges $\mathbf{e}$. We refer to this model as MAS-ValiGraph.

**Recurrent State Noise Model:** Letting $\mathbf{S} = \mathbf{T} = \mathbf{1}\mathbf{p}_e$ produces a recurrent state noise model with $d_e$ recurrent states. As in the multi-state absorbing noise model, we let the transition probability to each recurrent state be proportional to the empirical marginal probability of the specific edge type. However, we also enable transitions between the recurrent states. $\mathbf{P}_e$ is in this case shown to be idempotent analogously to the Multi-state Absorbing Noise model by exploiting that $\mathbf{T} = \mathbf{S} = \mathbf{1}\mathbf{p}_e\mathbf{1}\mathbf{p}_e$. Again, we have the limit distribution $\mathbf{e}\mathbf{P}_e = \mathbf{p}_e$. We refer to this model as RS-ValiGraph.

**Sampling Using the Reverse Process:** The sampling of new graphs is iteratively done by using $p_\theta(G^{t-1} \mid G^t)$ parametrized using a graph neural network. The exact construction of this process can be found in Appendix C. Remember that the forward model as applying an edge-deletion process by moving edges from an edge-state to an absorbing state, and an edge-addition process by moving edges from the no-edge state to the absorbing state. The reverse process should be interpreted in a similar light: An edge-addition process moving edges from the absorbing state to an edge-state, and an edge-deletion process (or edge-blocking process), where edges are moved from an absorbing state to the no-edge state.

When sampling graphs from $p_\theta(G^{t-1} \mid G^t)$ we importantly are not guaranteed that the desired properties are respected at each step. To ensure that this is the case we only move edges from the absorbing state to an edge/no-edge state, if we do not violate the edge-deletion invariant and edge-addition invariant properties respectively. This operation can be thought of as a projection onto the set of valid graphs (Madeira et al., 2024), and ensures that graphs have the correct structure during the intermediate steps. If an edge in an absorbing state cannot transition into an edge-state without violating structural constraints, it *must* be added to the no-edge state, and vice versa. As a consequence of this, it is theoretically possible for an edge in an absorbing state to be unable to transition to an edge- or no-edge state without violating structural constraints. We refer to this phenomenon as the graph generation procedure not converging: the process will not produce a graph. This can occur when the imposed constraints are severe, e.g. enforcing a graph being connected and disconnected will naturally not yield any graphs.

This validity check can be very computationally expensive. However, as the model only move edges from the absorbing state to an edge/non-edge state, we know that the graph at $G^{t-1}$ respects the structural constraints a priori. Verifying that the structural property in question is not violated can often be done at a lower computational cost.

## 4 EVALUATION: CAPTURING TOPOLOGICAL INFORMATION

Enforcing that the generated graphs lie within the support of the empirical distribution graph distribution can be done based on their structural properties, but is not alone sufficient for ensuring a good generative model. We therefore need to define appropriate metrics for evaluating the quality of a graph generative model.

**Conventional Evaluation of Structural Properties:** Often the Maximum Mean Discrepancy (MMD) (Gretton et al., 2012) computed using relevant graph features is used for evaluating the quality of the generative model. This approach was first used by (You et al., 2018) specifically using

the degree distribution, the clustering coefficient distribution, and average orbit counter, that is, how many times orbits which have exactly 4 nodes occur, as the graph statistics of interest. These statistics, occasionally supplemented by the eigenvalues of the normalized graph Laplacian (Liao et al., 2019), have become the de facto standard statistics to be used for computing the MMD for graph generative models (Eijkelboom et al., 2024; Vignac et al., 2022; Madeira et al., 2024; Martinkus et al., 2022; Diamant et al., 2023; Jo et al., 2022). However, it is not clear if they are sufficient, if the goal is to quantify higher order topological information. For example, a large linear graph and a large circular graph have very similar degree distributions, clustering coefficients and orbit counts, but they are very different topologically.

**Evaluation Using Extended Persistence Diagrams:**    We propose to supplement the conventional metrics with the the computation of MMD based on extended persistence diagrams to better quantify the ability of the models to capture topological information. Computations are based on the `Gudhi` library[1].

First, note that extended persistence diagrams are normally extracted from *simplicial complexes*. However, a graph can be represented as a 1-dimensional simplicial complex consisting of zero-simplices (the nodes), and one-simplices (the edges). Secondly, in order to compute extended persistence, one needs to define a *filtration* of the complex, that is, a sequence of nested simplicial complexes, whose final complex is the input simplicial complex itself. We do this by defining a real-valued function on the graph nodes $f : V \to \mathbb{R}$, and then considering a sequence of growing subgraphs, where each subgraph $G_\alpha$ contains only those vertices whose function values are less than an increasing threshold $\alpha$:

$$G_{\alpha_1} \subseteq G_{\alpha_2} \subseteq \cdots \subseteq G_{\alpha_n},$$

where $\alpha_1 \leq \alpha_2 \leq \cdots \leq \alpha_n$, and where $G_\alpha := \{V_\alpha, E_\alpha\}$, with $V_\alpha := \{v \in V : f(v) \leq \alpha\}$, $E_\alpha := \{(v, v') \in E : v, v' \in V_\alpha\}$. These subgraphs can be interpreted as the so-called *sub-level sets* of the function $f$, and can be used to compute ordinary persistence diagrams by tracking the threshold values of appearance and disappearance of topological features (connected components, branches, loops) in the sequence. In this work, we use the first eigenfunction of the graph Laplacian as the function $f$. This corresponds to a standard choice in topological data analysis applied to graphs, as the graph Laplacian is known to be related to the topological descriptors of a graph, e.g. the graph's homology groups. To construct an *extended* persistence diagram (Cohen-Steiner et al., 2009), we supplement the ordinary sequence of sub-level graphs with the sequence of super-level graphs. One advantage of extending the filtration is that disappearance times can be defined for graph loops and connected components by tracking the super-level graphs for which the features appear again, and using the corresponding $\alpha$ values as the disappearance times. See Appendix D for details on the construction of extended persistence diagrams.

**Interpretation of Persistence Diagrams**    When interpreting ordinary persistence diagrams, one challenge is that topological features can remain undetected, or depicted with infinite lifetimes, or persistence (i.e. the difference between disappearance and appearance times is infinite). This is addressed by using extended persistence diagrams. Depending on whether the appearance and disappearance times come from the sub- or the super-level graph sequence, extended persistence diagrams can have four types, each of which tracks the appearance/disappearance of specific topological features with respect to the filtration. $\mathbf{Ord}_0$ tracks branches pointing downwards based on the sub-level graphs. $\mathbf{Rel}_1$ tracks branches pointing upwards based on the super-level graphs. $\mathbf{Ext}_0^+$ tracks connected components from the sub-level and super-level graphs. $\mathbf{Ext}_1^-$ tracks loops based on the sub-level and super-level graphs.

Through persistent homology we thus obtain a method for quantifying the presence of branches – a feature which is hard to otherwise formalize. Also, instead of for instance merely counting loops and connected components, we gather information, not only about the presence of such features, but also on how "large" they are under the filtration.

**Choosing a Suitable Kernel**    Using extended persistence diagrams for MMD computation requires the choice of a suitable kernel defined on persistence diagrams. Our kernel of choice is the sliced

---

[1] https://gudhi.inria.fr/python/latest/

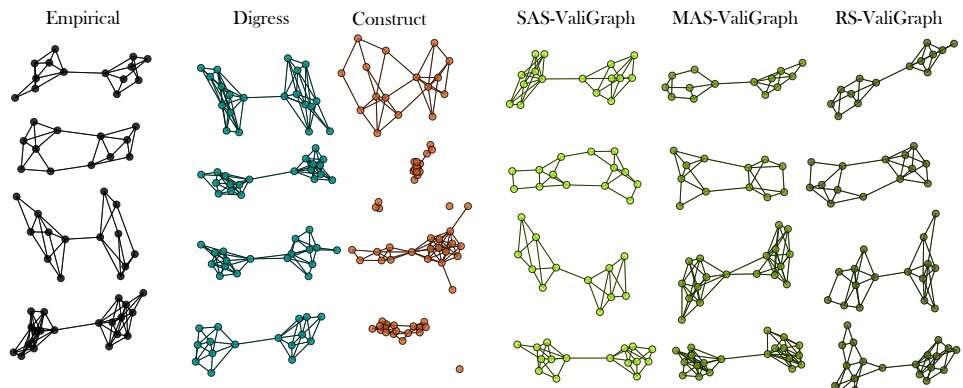

Figure 4: Graphs sampled from models trained on the Community dataset. We clearly see, how ValiGraph ensures the generation of connected graphs as opposed to ConStruct. Digressexhibits a similar ability to respect the structural constraint on this dataset.

Wasserstein kernel (Carrière et al., 2017), which is known to be theoretically both stable and discriminative. Using the fact that the sliced Wasserstein distance (Bonneel et al., 2015) is a valid distance between probability measures, we know from Berg et al. (1984), that a kernel can be defined as:

$$k_{\text{SW}}(D_1, D_2) := \exp\left(-\frac{\text{SW}(D_1, D_2)}{\sigma}\right), \tag{8}$$

where $D_1$ and $D_2$ are persistence diagrams (seen as probability measures), and $\text{SW}(D_1, D_2)$ refers to the sliced Wasserstein distance between these diagrams. This kernel is the Sliced Wasserstein Kernel, with bandwidth $\sigma$ as hyperparameter. A too large bandwidth can result in a too smooth kernel (large off-diagonal elements), and a too small bandwidth can result in a noise kernel (low off-diagonal elements). Hence, we set the bandwidth to be the median of the Sliced Wasserstein distances between all pairs of persistence diagrams extracted from the training dataset, see Appendix E.2 for details.

## 5 EXPERIMENTS

We evaluate the proposed ValiGraph with three different noise models: Single Absorbing State (SAS-ValiGraph), Multiple Absorbing States (MAS-ValiGraph), and Multiple Recurrent States (RS-ValiGraph). All the models are designed using the same Graph Transformer backbone (Dwivedi & Bresson, 2021). See Appendix C.1 for implementation and training details. We also implement versions of Digress (Vignac et al., 2022), and ConStruct (Madeira et al., 2024) following the authors' instructions, as they are comparable state-of-the art models for graph generation using discrete diffusion.

**Datasets**    To showcase ValiGraph, we consider three synthetic datasets with clear structural properties: connectedness, planarity, and tree structure. *Community* (You et al., 2018) refers to a small dataset of graphs drawn from a stochastic block model. All graphs of this dataset are connected, a structural criterion which we require to be respected for this dataset. *Planar* (Martinkus et al., 2022) is a synthetic dataset of planar and connected graphs, and finally *Lobster* (Liao et al., 2019) is a dataset of tree graphs, where each node is at most 2 hops away from a backbone. A valid lobster-graph has to be a tree (connected and with no cycles), and to reduce to linear when stripped of all nodes that are two hops away from leaf nodes. We provide statistics about the datasets in Appendix F.

**Enforcing Structural Constraints with ValiGraph:**    We first validate ValiGraph's ability to generate structurally valid graphs, see Table 1. For each dataset, we evaluate the proportion of the generated graphs which respect the relevant structural constraints for each dataset. If all constraints are respected for a graph, the graph is considered *valid*. If it is different from all other generated graphs up to isomorphism, it is *unique*, and if the generated graph is different from all graphs in the training dataset up to isomorphism, then we consider it *novel*. Graphs respecting all these properties being valid, unique, and novel (VUN) are also displayed. We find that variations of

Table 1: Results on Validity, Uniqueness, Novelty (VUN). We see, that ValiGraph consistently produces graphs with high validity while still having a varied output.

| | | Connected ↑ | Planar ↑ | Tree ↑ | Lobster ↑ | Valid ↑ | Unique ↑ | Novel ↑ | VUN ↑ |
|---|---|---|---|---|---|---|---|---|---|
| Community | Digress | 0.98 | - | - | - | 0.98 | 0.18 | 0.10 | 0.09 |
| | ConStruct | 0.60 | - | - | - | 0.60 | **0.97** | **1.00** | 0.60 |
| | SAS-ValiGraph | **1.00** | - | - | - | **1.00** | 0.85 | 0.96 | 0.83 |
| | MAS-ValiGraph | **1.00** | - | - | - | **1.00** | 0.75 | 0.85 | 0.72 |
| | RS-ValiGraph | **1.00** | - | - | - | **1.00** | 0.89 | 0.93 | **0.88** |
| Planar | Digress | **1.00** | 0.44 | - | - | 0.44 | **1.00** | **1.00** | 0.44 |
| | ConStruct | **1.00** | **1.00** | - | - | **1.00** | **1.00** | **1.00** | **1.00** |
| | SAS-ValiGraph | **1.00** | **1.00** | - | - | **1.00** | **1.00** | **1.00** | **1.00** |
| | MAS-ValiGraph | **1.00** | **1.00** | - | - | **1.00** | **1.00** | **1.00** | **1.00** |
| | RS-ValiGraph | **1.00** | **1.00** | - | - | **1.00** | **1.00** | **1.00** | **1.00** |
| Lobster | Digress | 0.92 | **1.00** | 0.84 | 0.67 | 0.67 | **1.00** | **0.98** | 0.65 |
| | ConStruct | 0.90 | **1.00** | **1.00** | 0.90 | 0.90 | **1.00** | 0.97 | 0.87 |
| | SAS-ValiGraph | **1.00** | **1.00** | **1.00** | 0.97 | 0.97 | **1.00** | **0.98** | **0.96** |
| | MAS-ValiGraph | **1.00** | **1.00** | **1.00** | **0.99** | **0.99** | **1.00** | 0.96 | 0.95 |
| | RS-ValiGraph | **1.00** | **1.00** | **1.00** | 0.97 | 0.97 | 0.98 | 0.97 | 0.93 |

Table 2: $\text{MMD}^2$ based on various extracted graph features. We see, that ValiGraph even with the imposed constraints is able to fit the graph distributions well.

| | | Spectral ↓ | Degree distribution ↓ | Orbit count ↓ | Clustering coefficient ↓ | $\text{Ord}_0$ ↓ | $\text{Rel}_1$ ↓ | $\text{Ext}_0^+$ ↓ | $\text{Ext}_1^-$ ↓ | Mean $\text{MMD}^2$ |
|---|---|---|---|---|---|---|---|---|---|---|
| Community | Digress | **0.045** | 0.042 | 0.077 | **0.174** | **0.176** | 0.153 | 0.113 | **0.109** | **0.138** |
| | ConStruct | 0.130 | 0.042 | 0.110 | 0.246 | 0.239 | 0.111 | 0.436 | 0.208 | 0.248 |
| | SAS-ValiGraph | 0.086 | **0.013** | **0.065** | 0.251 | 0.216 | 0.091 | 0.180 | 0.188 | 0.169 |
| | MAS-ValiGraph | 0.091 | 0.032 | 0.121 | 0.226 | 0.262 | **0.067** | **0.110** | 0.173 | 0.153 |
| | RS-ValiGraph | 0.121 | **0.013** | 0.066 | 0.239 | 0.196 | 0.123 | 0.186 | 0.217 | 0.180 |
| Planar | Digress | **0.010** | **0.002** | 0.024 | 0.126 | 0.071 | 0.048 | **0.048** | **0.042** | 0.052 |
| | ConStruct | 0.016 | **0.002** | 0.086 | **0.093** | **0.027** | **0.030** | 0.023 | 0.065 | **0.036** |
| | SAS-ValiGraph | 0.022 | 0.010 | 0.031 | 0.229 | 0.109 | 0.119 | 0.039 | 0.065 | 0.083 |
| | MAS-ValiGraph | 0.016 | 0.012 | 0.026 | 0.246 | 0.097 | 0.101 | 0.089 | 0.061 | 0.087 |
| | RS-ValiGraph | 0.020 | 0.012 | **0.020** | 0.248 | 0.149 | 0.103 | 0.038 | 0.058 | 0.087 |
| Lobster | Digress | **0.003** | **0.001** | 0.017 | **0.000** | 0.024 | 0.015 | 0.026 | 0.026 | 0.023 |
| | ConStruct | 0.004 | **0.001** | 0.006 | **0.000** | 0.020 | 0.021 | 0.029 | **0.000** | 0.017 |
| | SAS-ValiGraph | **0.003** | **0.001** | **0.011** | **0.000** | 0.023 | 0.019 | 0.020 | **0.000** | 0.016 |
| | MAS-ValiGraph | 0.004 | **0.001** | 0.014 | **0.000** | 0.028 | 0.015 | 0.014 | **0.000** | **0.014** |
| | RS-ValiGraph | 0.005 | **0.001** | 0.017 | **0.000** | 0.032 | **0.014** | **0.011** | **0.000** | **0.014** |

ValiGraph outperform, or perform on par with, other models in terms of VUN on all datasets. As expected, ConStruct preserves properties invariant to edge deletion (planarity and alicyclic); however, it fails to preserve the edge-addition invariant property of connectedness on Community and Lobster. ConStruct can be modified to preserve properties invariant to edge addition, by modeling the graph complement of a dataset or by considering a noise model invariant to edge addition. However, in doing so it would no longer guarantee properties which are invariant to edge deletion.

Digress ensures a very high validity when modeling the Community dataset which is consistent with the samples shown in 4. However, inspecting values for uniqueness and novelty, we see that the performance is caused by only generating a few graphs mostly from the training dataset, suggesting that the model does not generalize well in this specific case. This also explains On the Planar we see very poor performance in terms of generating planar graphs, highlighting the strong need to impose hard constraints as an inductive bias on the reverse process.

All versions of ValiGraph outperform the baseline methods in term of VUN on all datasets. Not one noise model outperforms on all datasets though, and choosing a suitable one may depend on the task at hand. Notably, even though ValiGraph enables us to enforce the generation of valid graphs throughout the reverse process, it does *not* generate all valid Lobster graphs. This is due to the fact, that there may be situations, where an edge, due to harsh property constraints, cannot transition from an absorbing state to an edge state or a non-edge state, and as a consequence the reverse process does not converge.

**Fitting the graph distributions:** A guarantee that the generated graphs are indeed valid is not sufficient for model evaluation, as a model which guarantees generation of valid graphs can achieve high uniqueness and novelty even if it does not fit the data distribution well. Thus, we also evaluate models' ability to fit the graph distribution using MMD based on conventional graph features (spectrum, degree distribution, orbit count, and clustering coefficient) as well as the MMD based on

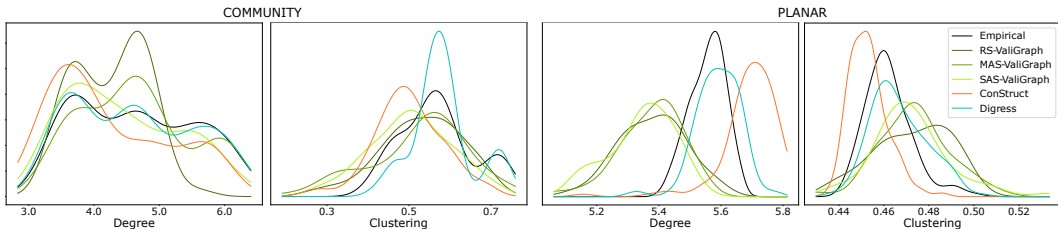

Figure 5: Kernel density plots of average node degree distribution, and average clustering coefficient.

persistence diagrams illustrating the models' ability to capture the topological features of the graphs in the empirical graph distribution. The $\mathrm{MMD}^2$ is calculated between features extracted from a set of generated graphs, and graphs from the test set. The results can be found in Table 2.

For the Community dataset we see, that Digress performs very well on all metrics. However, as Digress fails to generate many novel and unique molecules the good performance is to be expected. Most notably we see, that the proposed ValiGraph outperforms ConStruct on all metrics. Here in particular the MMD related to $\mathrm{Ext0}^+$, which relates to connected components, is reflects the inability to consistently generate connected graphs. When the conventional metrics are observed for all models fitted on the Lobster dataset, one could get the impression that the fitted models perform very similarly. However, again by $\mathrm{Ext}_0^+$ quantifies the inability of Digress and ConStruct to consistently produce connected graphs, and $\mathrm{Ext}_1^-$ reflects that Digress occasionally produces graphs containing cycles. Notably, on this dataset with very strict structural constraints ValiGraph consistently outperform the baseline methods on all metrics.

Disappointingly ConStruct performs better than ValiGraph in terms of fitting the data distribution on the Planar dataset. When also remembering that both methods produce all valid graphs, there should logically be a preference for ConStruct on this dataset. It seems, that as the Planar consists of very dense graphs there is no need for explicitly enforcing connectedness. The lower performance may also be caused by the fact that since ValiGraph blocks edges from being added not enough edges are added in the end. This explanation is consistent with 5 we see that ValiGraph generates graphs with lower average node degree than found empirically.

## 6 LIMITATIONS

There is a possibility that during generation not all edges are removed from the absorbing states, as it may be impossible for it to transition into the edge-state or the no-edge state without violating structural constraints. For some pairs of properties, for instance connectedness and tree-likeness, this is not a problem, as it is always possible to connect two tree graphs thus making a new tree, however, enforcing connectedness and disconnectedness simultaneously is obviously infeasible. The practical implications of this limitation has to be evaluated based on the application at hand.

## 7 CONCLUSION

In this paper we propose a recurrent state noise model, the use of which in discrete graph diffusion facilitates the construction of a model that is able to enforce hard structural constraints on the generated graphs, both invariant to edge-addition and edge deletion while still admitting a non-trivial limit distribution. We find empirically that the model does indeed manage to preserve these properties while still performing on par with comparable baseline models. Lastly, we suggest using the the Sliced Wasserstein kernel to compute maximum mean discrepancy based on extended persistence diagrams for evaluation the ability of graph generative models to capture information about graph topology, and demonstrate that conventional evaluation metrics do not necessarily capture this same information.

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

## A    COMPUTATION OF $\bar{\mathbf{Q}}^t$

Let $\mathbf{P} \in \mathbf{R}^{n \times n}$. $\mathbf{P}$ is called *idempotent* if $\mathbf{P} = \mathbf{P}^2$. Furthermore we will refer to $\mathbf{P}$ as a *stochastic matrix* if:

$$\mathbf{P}[i,j] \geq 0 \quad \text{for all } i,j \in \{1,...,n\} \text{ and}$$

$$\sum_{j=1}^{n} \mathbf{P}[i,j] = 1 \quad \text{for all } i \in \{1,...,n\},$$

that is, all entries are non-negative, and each row of the matrix sums to 1.

Let $\mathbf{P}$ be an idempotent stochastic matrix. Then $\mathbf{Q}^t = \alpha^t \mathbf{I} + (1 - \alpha^t)\mathbf{P}$, where $\mathbf{I}$ denotes an $n \times n$ identity matrix and $0 \leq \alpha^t \leq 1$ for $t \in \mathbb{N}$, is also a stochastic matrix, and $\bar{\mathbf{Q}}^t = \mathbf{Q}^1, ..., \mathbf{Q}^t$ can be written as:

$$\bar{\mathbf{Q}}^t = \bar{\alpha}^t \mathbf{I} + (1 - \bar{\alpha}^t)\mathbf{P}, \tag{9}$$

where $\bar{\alpha}^t = \prod_{k=1}^{t} \alpha^k$.

First we show that $\mathbf{Q}^t$ is a stochastic matrix. For all $i,j \in \{1,...,n\}$ and any $\alpha^t$, we have that $\mathbf{Q}^t[i,j] = \alpha^t + (1 - \alpha^t)\mathbf{P}[i,j] \geq 0$ as all terms are greater than 0.

The second assertion can be proven by induction over $t$. Now assume $t = 1$. Then we clearly see that:

$$\bar{\mathbf{Q}}^1 = \mathbf{Q}^1 = \alpha^1 \mathbf{I} + (1 - \alpha^1)\mathbf{P} = \bar{\alpha}^1 \mathbf{I} + (1 - \bar{\alpha}^1)\mathbf{P} \tag{10}$$

since $\alpha^1 = \bar{\alpha}^1$ by definition. For the induction step we pick $t \in \mathbb{N}$, and assume that the assertion holds for this choice of $t$. Then:

$$\bar{\mathbf{Q}}^{t+1} = \bar{\mathbf{Q}}^t \mathbf{Q}^{t+1}$$
$$= (\bar{\alpha}^t \mathbf{I} + (1 - \bar{\alpha}^t)\mathbf{P})(\alpha^{t+1}\mathbf{I} + (1 - \alpha^{t+1})\mathbf{P})$$
$$= \bar{\alpha}^t \alpha^{t+1} \mathbf{I} + \bar{\alpha}^t(1 - \alpha^{t+1})\mathbf{P} + \alpha^{t+1}(1 - \bar{\alpha}^t)\mathbf{P} + (1 - \bar{\alpha}^t)(1 - \alpha^{t+1})\mathbf{P}^2$$
$$= \bar{\alpha}^{t+1}\mathbf{I} + \beta^{t+1}\mathbf{P},$$

where we in the last step exploits that $\mathbf{P}$ is idempotent, and let:

$$\beta^{t+1} = \bar{\alpha}^t(1 - \alpha^{t+1}) + \alpha^{t+1}(1 - \bar{\alpha}^t) + (1 - \alpha^{t+1})(1 - \bar{\alpha}^t)$$
$$= \bar{\alpha}^t(1 - \alpha^{t+1}) + (1 - \bar{\alpha}^t)$$
$$= \bar{\alpha}^t - \bar{\alpha}^{t+1} + 1 - \bar{\alpha}^t)$$
$$= 1 - \bar{\alpha}^{t+1}.$$

The assertion now follows from the principle of induction.

## B    NOISE SCHEDULES

To enable efficient computation of transition matrices at training time, while still ensuring low memory consumption we pre-compute $\mathbf{P}$ as well as $\alpha^1, ..., \alpha^T$ and $\bar{\alpha}^1, ..., \bar{\alpha}^T$ respectively, and save these in memory, before training commences. Using this information we can conveniently construct transition matrices $\mathbf{Q}^t$ and $\bar{\mathbf{Q}}^t$ instead of having to pre-compute, and store these in memory. In the following we will specify the exact formulation of the noise schedules utilized in this paper. The utilized noise schedules are shown in Figure 6.

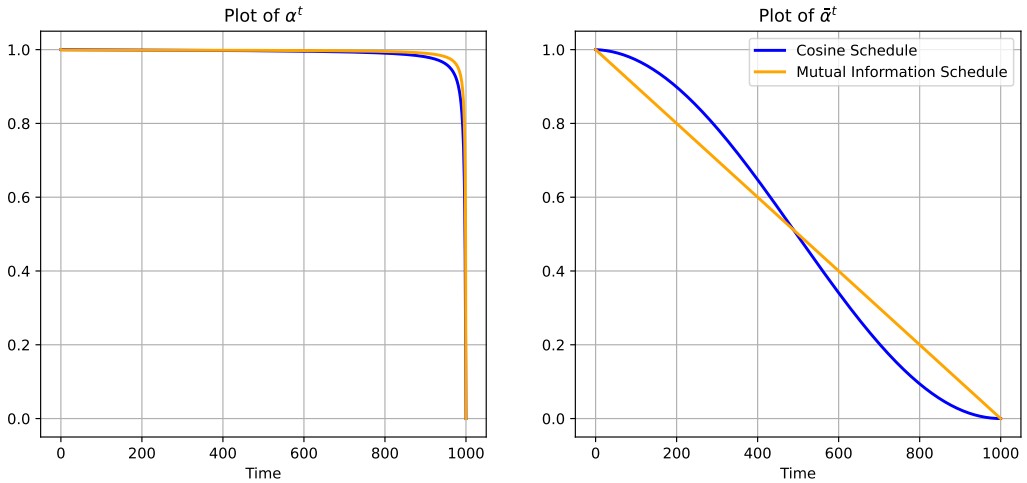

Figure 6: The depiction of $\alpha^t$ (left) and $\bar{\alpha}^t$ (right) for the Cosine Noise Schedule, and the Mutual Information Noise Schedule respectively. Both of them are computed for $T = 1000$.

### B.1    COSINE SCHEDULING

The de facto standard noise-schedule for diffusion models is the popular cosine schedule of Nichol & Dhariwal (2021). Given a maximum number of diffusion steps $T$, we define a function $f : \{0, ..., T\} \rightarrow \mathbb{R}$ as:

$$f(t) = \cos\left(\frac{\pi}{2}\frac{\left(\frac{t}{T} + s\right)}{(1 + s)}\right)^2, \tag{11}$$

where $s$ is a small constant to be considered a hyperparameter. We choose $s = 0.008$ consistent with (Nichol & Dhariwal, 2021; Madeira et al., 2024; Vignac et al., 2022). Now we define:

$$\bar{\alpha}^t = \frac{f(t)}{f(0)}. \tag{12}$$

Notice, that $\bar{\alpha}^0 = 1$ by construction. By remembering that $\bar{\alpha}^t = \prod_{k=0}^{t} \alpha^k$ we can now define $\alpha^t$ as:

$$\alpha^t = \begin{cases} \frac{\bar{\alpha}^t}{\bar{\alpha}^{t-1}} & \text{if } t = 1, ..., T \\ \bar{\alpha}^t & \text{if } t = 0 \end{cases}. \tag{13}$$

This noise schedule is what is referred to as the *Cosine Schedule*.

### B.2    MUTUAL INFORMATION

The mutual information schedule (Sohl-Dickstein et al., 2015; Austin et al., 2021) can be interpreted as linearly increasing the probability of being in an absorbing state over time. That is we arrive at the definition by desiring a noise schedule which erases a constant fraction of our signal each each diffusion step. That is, given a maximum number of diffusion steps $T$ we aim for:

$$\bar{\alpha}^t = 1 - \frac{t}{T}, \tag{14}$$

for $t \in \{0, ..., T\}$. Again, we have that $\bar{\alpha}^0 = 1$ by construction, and we can define $\alpha^t$ analogously to the Cosine Schedule, that is:

$$\alpha^t = \begin{cases} \frac{\bar{\alpha}^t}{\bar{\alpha}^{t-1}} & \text{if } t = 1, ..., T \\ \bar{\alpha}^t & \text{if } t = 0 \end{cases}. \tag{15}$$

Here, we see that:

$$\frac{\bar{\alpha}^t}{\bar{\alpha}^{t-1}} = \frac{1 - \frac{t}{T}}{1 - \frac{t-1}{T}} = \frac{T - t}{T - t + 1} = 1 - \frac{1}{T - t + 1} \tag{16}$$

Which is consistent with the definition of Sohl-Dickstein et al. (2015). We refer to this construction as the *Mutual Information* noise schedule.

## C    REVERSE PROCESS

The construction of the reverse process follows the general process of Vignac et al. (2022). We include information about it here for the sake of completeness.

### C.1    TRAINING REGIME AND ARCHITECTURE

We train a denoising neural network predicting the probabilities of each node- and edge, i.e. $\hat{p}^G = (\hat{p}^X, \hat{p}^E)$ at time-step zero type given a noisy input. This is done using a standard cross-entropy loss, i.e.:

$$L(\hat{p}^G, G) = CE(\hat{p}^X, \mathbf{X} + \lambda CE(\hat{p}^X, \mathbf{E})), \tag{17}$$

where $\lambda$ is a hyperparameter governing the trade-off between node- and edge-losses. However, as we do not consider datasets with node-attributes this hyperparameter does not affect the outcomes of the experiments. Importantly, we need to choose a permutation equivariant architecture to ensure that this loss is invariant under permutation, which motivates the choice of Graph Neural Network, in particular a Graph Transformer (Dwivedi & Bresson, 2021). To ensure fair comparison of the models in question we use the same backbone and hyperparameters as ConStruct(Madeira et al., 2024), with the one exception, that additional graph features used to augment the noisy graph input to increase the expressivity of the architecture, is not only computed on the graphs constructed throughout the edge-addition denoising process, but also on the ones constructed during the edge-deletion denoising process.

For the implementation of the baseline models i.e. ConStruct and Digress we use the official implementation provided by the authors with minimal changes applied to adapt these to our specific setup. The models were then trained using the configuration provided by the authors. Relevant code will be made available upon publication containing configuration files for each model.

## C.2 Construction of the Reverse Process

Consistent with the noising process we model the denoising process $p_\theta(G^{t-1} \mid G^t)$ using the assumption of independent sampling of nodes and edges as:

$$p_\theta(G^{t-1} \mid G^t) = \prod_{i\in\{1,\dots,n\}} p_\theta(\mathbf{x}_i^{t-1} \mid G^t) \cdot \prod_{i,j\in\{1,\dots,n\}} p_\theta(\mathbf{e}_{ij}^{t-1} \mid G^t) \tag{18}$$

Each term can then be computed by marginalizing over the network predictions. That is:

$$p_\theta(\mathbf{x}_i^{t-1} \mid G^t) = \sum_{d\in\{1,\dots,d_x\}} p_\theta(\mathbf{x}_i^{t-1} \mid \mathbf{x}_i[d] = 1, G^t)\hat{p}_i^X[d] \tag{19}$$

where $\hat{p}_i^X[d]$ denotes the prediction of the networks predicted probability of node $i$ being of type $d$, and where we choose to model $p_\theta(\mathbf{x}_i^{t-1} \mid \mathbf{x}_i[d] = 1, G^t)$ as:

$$p_\theta(\mathbf{x}_i^{t-1} \mid \mathbf{x}_i[d] = 1, G^t) = \begin{cases} q(\mathbf{x}_i^{t-1} \mid \mathbf{x}_i[d] = 1, \mathbf{x}_i^t) & \text{when } q(\mathbf{x}_i^t \mid \mathbf{x}_i[d] = 1) > 0 \\ 0 & \text{otherwise.} \end{cases} \tag{20}$$

where the posterior $q(\mathbf{x}_i^{t-1} \mid \mathbf{x}_i[d] = 1, \mathbf{x}_i^t)$ is given as:

$$q(\mathbf{x}_i^{t-1} \mid \mathbf{x}_i[d] = 1, \mathbf{x}_i^t) = \frac{\mathbf{x}_i^t(\mathbf{Q}_x^t)^\top \odot \mathbf{x}_i\bar{\mathbf{Q}}_x^{t-1}}{\mathbf{x}_i^t\bar{\mathbf{Q}}_x^t\mathbf{x}_i} \tag{21}$$

The terms $p_\theta(\mathbf{e}_{ij}^{t-1} \mid G^t)$ related to the distribution over edges can be computed in a similar manner.

## D Creating Persistence Diagrams

Strictly speaking, extended persistence diagrams need *simplicial complexes* as inputs. However, a graph can be represented as a 1-dimensional simplicial complex consisting of zero-simplices (the nodes), and one-simplices (the edges). Moreover, in order to compute extended persistence, one needs to define a *filtration* of the complex, that is, a sequence of nested simplicial complexes, whose final complex is the input simplicial complex itself. In our case, this can be done by defining a real-valued function on the graph nodes $f : V \to \mathbb{R}$, and then considering a sequence of growing subgraphs, where each subgraph $G_\alpha$ contains only those vertices whose function values are less than an increasing threshold $\alpha$:

$$G_{\alpha_1} \subseteq G_{\alpha_2} \subseteq \cdots \subseteq G_{\alpha_n},$$

where $\alpha_1 \leq \alpha_2 \leq \cdots \leq \alpha_n$, and where $G_\alpha := \{V_\alpha, E_\alpha\}$, with $V_\alpha := \{v \in V : f(v) \leq \alpha\}$, $E_\alpha := \{(v, v') \in E : v, v' \in V_\alpha\}$. Note that these subgraphs can be interpreted as the so-called *sub-level sets* of the function $f$. In this article, we use the first eigenfunction of the graph Laplacian as the function $f$. This corresponds to a standard choice in topological data analysis applied to graphs, as graph Laplacians are known to be related to the topological description of the graphs, such as their homology groups.

After having defined such a sequence, persistence diagrams can be computed by tracking the threshold values of appearance and disappearance of topological features (connected components, branches, loops) in the sequence. For instance, branches pointing downwards w.r.t. the filtration values can be detected as connected components of subgraphs that appear at a threshold $\alpha_b$, and then get merged at threshold $\alpha_d \geq \alpha_b$. The resulting interval $[\alpha_b, \alpha_d]$ can be turned into a point in $\mathbb{R}^2$. As $\alpha_b \leq \alpha_d$ this point is naturally located above the diagonal, and the collection of such points is the so-called *ordinary persistence diagram*. The values $\alpha_d - \alpha_b$ is often referred to as the *lifetime*, or *persistence*, of the topological feature.

When interpreting ordinary persistence diagrams, one challenge is that some topological features remain undetected (for instance, branches pointing upwards w.r.t. the filtration values), and some features are depicted with infinite persistence. For instance graph connected components never ends, and loops are never filled throughout the sequence of graphs, and as such these will have infinite persistence. To address this short-coming, we can instead use extended persistence (Cohen-Steiner et al., 2009), which complements the ordinary sequence of sub-level graphs with the sequence of super-level graphs.

One advantage of extending the filtration is that disappearance times can be defined for graph loops and connected components by tracking the super-level graphs for which the features reappear, and using the corresponding $\alpha$ values as the disappearance times. Moreover, branches pointing upwards, with respect to the filtration, can be detected in this sequence of super-level graphs, in the exact same way that downwards branches were in the sub-level graphs. Finally, depending on whether the appearance and disappearance times come from the sub- or the super-level graph sequence, extended persistence diagrams can have four types: $\text{Ord}_0$ (branches pointing downwards—both times come from the sub-level graphs), $\text{Rel}_1$ (branches pointing upwards—both times come from the super-level graphs), $\text{Ext}_0^+$ (connected components—appearance and disappearance times come from the sub-level and super-level graph sequences, respectively, and the point is above the diagonal), and $\text{Ext}_1^-$ (loops—appearance and disappearance times come from the sub-level and super-level graph sequences, respectively, and the point is below the diagonal). See Figure 7 for an illustration. See also (Carrière et al., 2019, Section 2.1) for a more complete description of the connections between extended persistence diagram points, and graph features such as connected components, branches and loops.

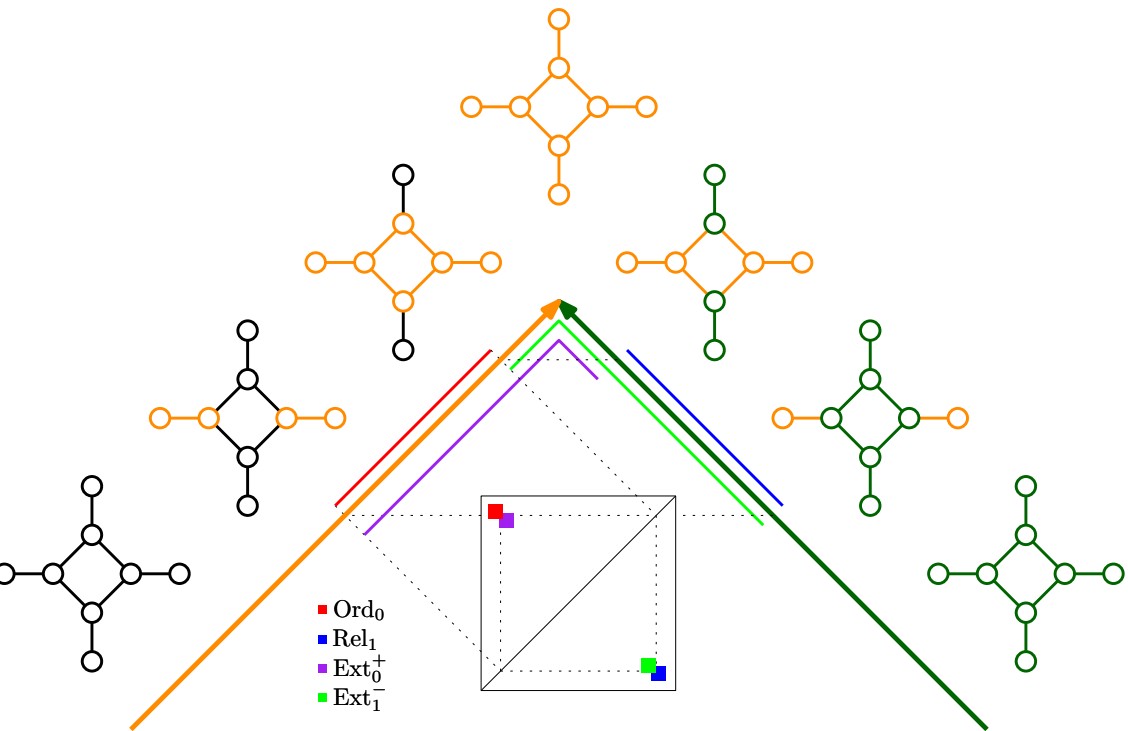

Figure 7: Example of extended persistence diagram computed on a graph. The sub-level graphs are displayed in orange, and the super-level graphs are displayed in dark green.

# E   EVALUATING GRAPH GENERATIVE MODELS

## E.1   THE MAXIMUM MEAN DISCREPANCY

For the evaluation of whether a graph generative model respects the topological properties of the distribution of graphs which it attempts to model, graph features are typically extracted, and the distribution of these features are then compared using *Maximum Mean Discrepancy* (MMD) as a proxy for comparing the graph distributions. To compute an empirical estimate of the squared MMD (Gretton et al., 2012) the following estimator is often chosen:

$$\widetilde{\text{MMD}}^2(x,y) = \frac{1}{n^2}\sum_{i=1}^{n}\sum_{j=1}^{n}k(x_i,x_j) + \frac{1}{m^2}\sum_{i=1}^{m}\sum_{j=1}^{m}k(y_i,y_j) - \frac{2}{nm}\sum_{i=1}^{n}\sum_{j=1}^{m}k(x_i,y_j), \quad (22)$$

where $k : \mathcal{G} \times \mathcal{G} \to \mathbb{R}$ is a suitable kernel and $n$ and $m$ denotes the number independent identically distributed samples from each distribution. This is however, as also stated by Gretton et al. (2012), a biased estimator of the $MMD^2$. To retrieve an unbiased estimator of the MMD one can exclude the diagonal kernel elements from the computation, and adjust the average accordingly. That is by choosing the estimator:

$$\widehat{\text{MMD}}^2 = \frac{1}{n(n-1)} \sum_{i=1}^{n} \sum_{j \neq i}^{n} k(x_i, x_j) + \frac{1}{m(m-1)} \sum_{i=1}^{m} \sum_{j \neq i}^{m} k(y_i, y_j) - \frac{2}{nm} \sum_{i=1}^{n} \sum_{j=1}^{m} k(x_i, y_j) \quad (23)$$

However, doing this, comes with the caveat, that the MMD estimate can be negative. In this paper we will follow the literature in the field and use the biased $\widetilde{MMD}^2$ of an estimator of the MMD. Additionally, also following the convention set by the literature in the field, we will refer to this estimate as MMD even though it is an estimate of the squared MMD.

### E.2 SLICED WASSERSTEIN KERNEL

Using extended persistence diagrams for MMD computation is not a trivial process. As we know, to compute the MMD estimate, a suitable kernel defined on persistence diagrams has to be chosen. In this paper we consider the Sliced Wasserstein kernel (Carrière et al., 2017), since it's metric properties are known to be equivalent to that of the original distances between persistence diagrams.

Given $\theta \in \mathbb{R}^2$ with $\|\theta\|_2 = 1$, we let $L(\theta)$ denote the line $\{\lambda\theta : \lambda \in \mathbb{R}\}$, and let $\pi_\theta : \mathbb{R}^2 \to L(\theta)$ be the orthogonal projection onto $L(\theta)$. Let $D_1, D_2$ be two persistence diagrams, and let $\mu_1^\theta := \sum_{p \in D_1} \delta_{\pi_\theta(p)}$ and $\mu_{1\Delta}^\theta := \sum_{p \in D_1} \delta_{\pi_\theta \circ \pi_\Delta(p)}$ be two sums of Dirac measures associated to $D_1$, and similarly for $\mu_2^\theta$, where $\pi_\Delta$ is the orthogonal projection onto the diagonal. Then, the *Sliced Wasserstein distance* is defined as:

$$\text{SW}(D_1, D_2) := \frac{1}{2\pi} \int_{\mathbb{S}_1} \mathcal{W}(\mu_1^\theta + \mu_{2\Delta}^\theta, \mu_2^\theta + \mu_{1\Delta}^\theta) \mathrm{d}\theta,$$

where $\mathcal{W}$ stands for the Wasserstein distance between probability measures. Hence, Berg et al. (1984) allows us to define a valid kernel with:

$$k_{\text{SW}}(D_1, D_2) := \exp\left(-\frac{\text{SW}(D_1, D_2)}{\sigma}\right).$$

Note that the Wasserstein distance can *not* be used directly in defining a kernel in an analogous way as the resulting kernel would not in general be positive semidefinite (PSD).

Evaluating the kernel $k_{\text{SW}}$ comes with a degree of uncertainty induced by the two chosen hyperparameters: the bandwidth $\sigma$ and the number of lines/slices.

If the bandwidth $\sigma$ is too large will result in a kernel which is too smooth, i.e. all off-diagonal elements are large, and a bandwidth which is too small will result in a noise kernel, i.e. all off-diagonal elements are small. Hence, we a reasonable choice is to set the bandwidth to be the median of the Sliced Wasserstein distances between all pairs of persistence diagrams.

The Sliced Wasserstein distance is defined as an integral over $\mathbb{S}_1$. In general this is not tractable, and instead we compute a Monte-Carlo estimate by sampling of $\theta$ over the interval $[-\pi/2, \pi/2]$, and evaluating the integrand with respect to the line $L(\theta)$. The number of lines/slices sampled in this estimate is the second hyper-parameter which must be set. At default this value is set to 10 to speed up computation, however, we use 100 directions to ensure as accurate a computation as possible within the computational budget.

## F DATASETS

For all experiments we use 80% of the available data as training data, 10% for validation and 10% for testing. Note, that the train-validation-test split is done at random and do not necessarily correspond to the ones used by in other papers, which may cause our results to deviate from the ones they report. However, as the same dataset splits are used for the training of the suggested models and the baseline models in our setup this is not a pose a problem for the interpretation of our results.

The Lobster is generated as 500 random lobster graphs using the lobster graph generator used by Liao et al. (2019) and Madeira et al. (2024). That is random lobster graphs are generated using the NetworkX library (Hagberg et al., 2008) with the properties:

- Maximum number of nodes: 100.
- Minimum number of nodes: 10.
- Mean number of nodes: 80.
- Probability of adding an edge to the backbone as well as the probability of adding an edge one level after the backbone: 0.7

The Planar dataset and the Community dataset are obtained from the implementation provided by Martinkus et al. (2022) to ensure consistency with ConStruct and Digress. The Planar dataset consists of planar graphs with exactly 64 nodes, and the number of edges pr. graph lying between 173 and 181. The Community dataset suggested by You et al. (2018) consists of 100 graphs drawn from a stochastic block model with two communities. The graphs have between 12 and 20 nodes each.

# G  GRAPH SAMPLES

In this section we show samples from the considered modes which – due to space constraints – were not included in the main paper. in Figure 8 samples from the Lobster dataset are shown, and in Figure 9 samples from the Planar dataset are shown.

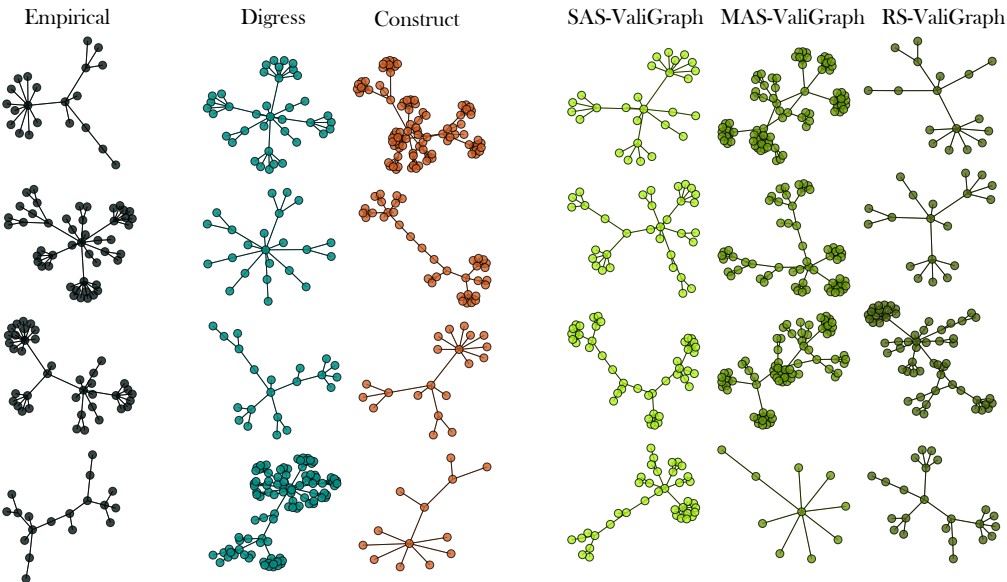

Figure 8: Lobster Graphs

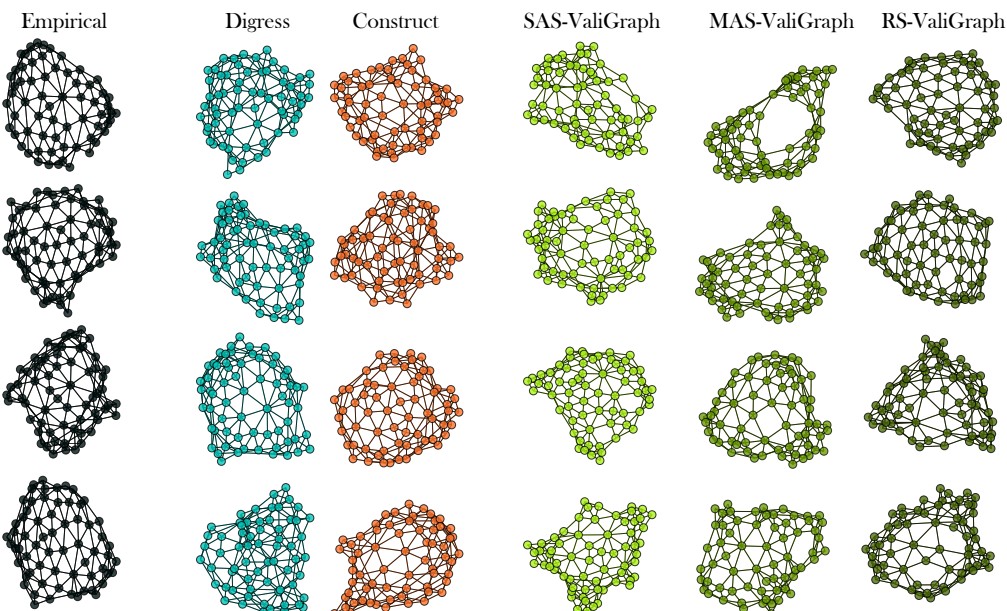

Figure 9: Planar Graphs

# H  DIFFUSION

Generative models based on denoising diffusion have proven to be extremely effective for image generation. This modeling paradigm is not only applicable to images, but can be used widely,

including for graph generation. A diffusion model consists of two main features: A noise model and a denoising neural network. For readers familiar with variational autoencoders (VAEs), the noise model is analogous to an encoder, and the denoising neural network is analogous to a decoder.

**The noise model:**   One main difference between the noise model and the encoder of a VAE is that it is not learned or estimated. This property is achieved by designing the noise model to have the following properties:

1. The noise model should have a **Markovian structure**, that is, $q(z^1, \ldots, z^T \mid z_0) = \prod_{t=1}^{T} q(z^t \mid z^{t-1})$, where $z_0 = x$.

2. Ensure that the noise model admits a **limit distribution that does not depend on** $x$. That is, $q(z^T \mid x)$ converges point-wise to a valid limit distribution $p(z)$, when $T \to \infty$. This is similar to requiring convergence in distribution. This limit distribution $p(z)$ will be used as the latent *prior*. As a consequence of this property, a sample from $q(z^T \mid x)$ can be thought of as a sample from $p(z)$ when $T$ is chosen sufficiently large.

3. Importantly, we require $q(z^t \mid x)$ to have a **closed form solution**, in order to ensure that it can be computed efficiently during training.

**Generative model:**   Letting $\theta$ denote the model parameters, the goal of the diffusion model is to sample from $p_\theta(x, z^1, \ldots, z^T) = p_\theta(x \mid z^1)p_\theta(z^1 \mid z^2) \cdots p_\theta(z^{T-1} \mid z^T)p(z^T)$, rather than from $p_\theta(x, z^T) = p_\theta(x \mid z^T)p(z)$, as in the VAE setting. However, to implement this efficiently, we need to be able to model $p_\theta(z^{t-1} \mid z^t)$. In theory, each distribution could be modeled separately; however, for large $T$ such an approach is not practically feasible. To overcome this, we model $p_\theta(z^{t-1} \mid z^t)$ by marginalizing over the network predictions:

$$p_\theta(z^{t-1} \mid z^t) = \int q(z^{t-1} \mid z^t, x)p_\theta(x \mid z^t)dx, \tag{24}$$

Where $p_\theta(z^{t-1} \mid z^t)$ should be available in closed form. Thanks to this formulation, we only need one model to estimate $p_\theta(x \mid z)$.

