# OpenReview forum: "But is it Valid? Enforcing Structural Constraints on Graph Generative Models"
_ICLR.cc/2026/Conference — Submitted to ICLR 2026_

### Official Review · Reviewer_Sfct · 2025-10-24

**Soundness:** 1
**Presentation:** 2
**Contribution:** 1
**Rating:** 2
**Confidence:** 4

**Summary:**

This paper proposes a generative model for graphs designed to explicitly enforce structural properties during generation. The framework is restricted to structural properties that are invariant under edge addition and edge deletion.

The paper argues that existing graph generative models often fail to capture such structural characteristics and introduces new metrics to evaluate this aspect. Several noising procedures are discussed, and the proposed model aims to preserve structural properties by only allowing edge transitions from an absorbing state to an edge or non-edge state when doing so does not violate the edge-addition or edge-deletion invariance conditions.

**Strengths:**

The paper is clearly written and relatively easy to follow.

The motivation of enforcing structural properties in graph generation is an important field of research.

**Weaknesses:**

### Scope and Applicability
The framework is limited to properties that can be preserved under edge deletion and edge addition. This restriction considerably narrows the range of structural properties the method can handle. It would be helpful for the authors to discuss how this limitation affects the general applicability of the model, and whether extensions to other classes of properties could be envisioned.

### Property Preservation Mechanism
The method operates by checking whether a given edge addition or deletion preserves the targeted property. However, the paper does not explain how this check is performed in practice. Providing a precise description of this verification step would improve both the clarity and reproducibility of the work.

### Noising Procedures
The noising procedures described in the method section are largely known from previous literature, with the exception of the proposed multi-absorbing-state procedure. However, the multi-absorbing-state noising process appears theoretically redundant, as it seems equivalent to using a single absorbing state. Clarifying the motivation and effect of this design choice would strengthen the technical contribution.

### Contextualization and Related Work
The paper would benefit from a more comprehensive discussion of related work. Many recent graph generative models are not cited, including GRUM, CatFlow, SID, and DeFog, as well as recent approaches for conditional or guidance-based generation. Expanding the literature review would help situate the proposed method within the current research landscape and highlight its specific novelty.

### Computational Cost
While computational complexity is mentioned, it is not empirically evaluated. Providing at least an estimate or a comparison of the computational overhead relative to existing baselines would make the discussion more complete and help assess the method’s practical usability.

**Questions:**

I do not have question.

See above form suggestions.

---

### Official Review · Reviewer_M2V7 · 2025-10-28

**Soundness:** 2
**Presentation:** 1
**Contribution:** 2
**Rating:** 2
**Confidence:** 2

**Summary:**

The paper introduces ValiGraph a method for discrete graph generation that accounts for structural constraints that the generated graphs should respect. The structural constraints and structures with which ValiGraph is able to deal are ones that are invariant to edge addition, e.g. community structures, or invariant to edge deletion, e.g. planarity.  Previous work, ConStruct [1], was preserving properties that are invariant to edge deletion. ValiGraph is instantiated with three different noise models: a single-state absorbing model (noise distribution is the Dirac delta on the absorbibg state), a multi-state absorbing model  and a recurrent state model (noise distribution is the marginal distribution of edge types). The method is evaluated on a set of synthetic datasets which have structural properties such as: connectivity, planarity, tree, and lobster structure (a particular type of tree structure), using standard measures of structural evaluation: such as validity, novelty, uniqueness, and standard measures of distributional alignement such MMD over the degree, clustering coefficient, average orbit counter and eigen values.  In addition to these measures they also introduce a set of measures that seek to capture higher order topological information using what are called extended persistence diagrams. The evaluation results show that the method improves performance with respect to the preservation of structural properties (validity, uniqueness, novelty) over the two baselines it compares against: DiGress (no structural constraints) ConStruct (only structural properties that are invariant to edge deletion). When to the distribution alignment the image is rather mixed, with Digress giving better alignment on Community, ConStruct on Planar, and ValiGraph on Lobster.

[1] Madeira, Manuel / Vignac, Clément / Thanou, Dorina / Frossard, Pascal Generative Modelling of Structurally Constrained Graphs 2024, NeurIPS 2024.

**Strengths:**

The adaptation of discrete diffusion so that it preserves structural properties that are invariant to edge addition and not only to edge deletion which has already been explored in previous work.

**Weaknesses:**

As described above the main contribution of the paper is being able to tackle also structural properties that are invariant to edge addition, which is somehow incremental with respect to previous work. This would have been fine if the presentation of the paper would have been clear and concise, but I had a hard time trying to parse parts of it, more on that bellow, in order to put the bigger picture together.

So the paper introduces some notation and machinery that I guess should relate to the structural property preservation. Since the paper focuses on edge deletions/additions it introduces noise models which are specific to the edges and which I guess should preserve the properties in question. For the edge representation this the paper defines, in addition to the various edge types that a given graph might have, the no-edge type/state, and auxiliary states. The latter can be one or more absorbing states, or recurrent states. The one-hot edge representations are extended to include the new types/states. For the one, more absorbing states, and recurrent states the paper defines appropriate noising processes by introducing the respective transition matrices which lead to different instantiations of the method.

In addition to the above the paper also introduces two edge projection operators which project from the edge types (including the no-edge) to a two dimensional binary representation:
* $\pi_+$ which seems to project an edge representation to a vector where
  * the first dimension indicates whether the edge is:  a non-edge or auxiliary state
  * and the second dimension indicates whether the edge is a standard edge (stupid question: why one would need two dimensions? unless the two dimensions are not mutually exclusive)
* $\pi_-$ which seems to project an edge representation to a vector where
  * the first dimension indicates whether the edge is a no-edge
  * and the second dimension indicates whether the state is absorbing or recurrent.

The above projections are used (not very rigorously in terms of notations, since the operators are defined on the level of edges but are now used over the graphs which have edges and nodes, what happens to the nodes?) to define a series of binary subgraphs by projecting noisy graphs $G_0$ (data) to $G_T$ (noise) produced during the noising process.  Somehow counterintuitively $\pi_+$ is used to denote an edge deletion process and $\pi_-$ to denote an edge addition process, but be it. Now I have to say that I fail to see the utility of the projection operator. The paper notes line 146: "Designing the noise model in this way has the remarkable consequence, that, during inference, we can guarantee any graph property which can be composed by both edge-deletion and edge-addition invariant properties." but so far no noise model has been defined, this comes later. So are the binary graphs really used? does this projection take place or it serves for demonstration purposes. How what is described here and visualised in figure 3 is really instantiated in the modelling process, noising/denoising?

More important, how the property preservation is guaranteed in the denoising step? In the ConStruct paper there has been quite some discussion on the use of a projection operator that is applied at every step of the denoising operation to make sure that the next step graph respects the structural properties, together also with some optimality guarantees. In the paper under review there is a vague reference to how edges can be moved from the absorbing state to the edge or no-edge state if they do not violate constraints, lines 245-248, but not much is said about how is this achieved for the different types of constraints considered, planarity, connectivity... In addition doing this at each denoising step has a computational cost which can be significant.

**Questions:**

In defining the noise models I needed some help to figure out that the single-state absorbing and Recurrent state noise models correspond to the masking, dirac distribution, noise model and the marginal edge distribution respectively (but probably this is my problem). Now why is there a need for a multi-state absorbing model, what does it bring with respect to the single absorbing state.

**Clarity questions**
1. Definition 3.1 seems to focus only on edges, i.e. $G' \subseteq G$ , but a graph is a set of nodes and a set of edges, so what happens when the node types change? as it can happen in the noising process. Then it is not obvious how one can define the subgraph relation because the graphs will not have the same nodes anymore.
2. Related to the above, lines 141-146: $\pi_{+/-}$ have been defined on edges but here are applied on the whole of the graphs. I understand this application as kind of returning a set of edge states (no edge/absorbing/recurrent state vs standard state for $\pi_+$, and no edge vs aborbing/recurrent state for $\pi_-$)  but I cannot get the semantics correctly, the notation is not very helpful. I also have the impression that in the definition of $\pi_+$ the $e_1$ (denoting edge presence/absence) should be added to the standard properties and not to the auxiliary ones, at least when I try to understand the operations as visualised in figure 3 and explained in terms of edge deletions and additions (e.g. "At each timestep, there is a chance of either edges- or non-edges transitioning to an absorbing state indicated by a gray edge.").
3. What is the relation of $\mathbf P_x$, lines 189-191, and $\mathbf p_x$ (eq. 5), is it that it is $\mathbf P_x = {\mathbf 1}_{d_x}{\mathbf p}_x$, I would take that to mean the outer product of the two vectors of dimension $d_X$ each.
4. In general I would have welcomed some more rigor in the notations, e.g. in the vector operations given two vectors $\mathbf 1$ and $\mathbf p$ then the inner product of the two could be denoted as $\mathbf 1^T \mathbf p$, while the outer product could be denoted as $\mathbf 1 \mathbf p^T$, would have make easier to parse a good number of parts of the paper.

---

### Official Review · Reviewer_akZ1 · 2025-10-31

**Soundness:** 2
**Presentation:** 2
**Contribution:** 3
**Rating:** 2
**Confidence:** 3

**Summary:**

This paper proposes ValiGraph, a diffusion-based graph generative model designed to produce graphs that strictly respect structural constraints such as connectedness, planarity, and tree structure. The model modifies the discrete denoising diffusion process to include absorbing and recurrent states that enforce edge-addition and edge-deletion invariants.
The authors also introduce an evaluation framework using extended persistent homology, arguing that conventional metrics like degree distribution or clustering coefficient fail to capture topological validity.
Experiments on synthetic datasets show that ValiGraph can generate valid, unique, and novel graphs that respect constraints better than existing diffusion-based models such as DiGress and ConStruct.

While the proposed ValiGraph is novel and interesting, some of its conclusions lack strong theoretical or empirical support. Specifically:

1. The paper mentions “Documenting that existing one-shot graph generative models struggle to capture topological properties of graph distributions” as one of this work's main contributions. But it doesn’t provide theoretical reasoning or comprehensive experimental evidence to back this up. Several one-shot models already aim to capture such properties — for instance, Zahirnia et al. (2022, 2023) show that defining log-likelihoods over structural features (like the k-step transition matrix) can effectively capture global graph structures in GVAE.
2. The proposed evaluation metric based on extended persistent homology is motivated by the idea that these metrics can better capture graph structural properties. However, (a) there is no clear theoretical or empirical demonstration showing that this metric correlates with meaningful structural aspects such as connectivity. (b) In addition, the paper does not discuss GNN-based evaluation metrics (see Thompson et al., 2022; Shirzad et al., 2022) or clarify whether, and to what extent, the proposed approach can better capture the topological properties of graphs compared to those methods.
For instance, it remains unclear how the proposed metric behaves under controlled perturbations of the ground-truth graphs, or how its sensitivity compares to GNN-based metrics in capturing deviations from the original graph structure.
3. The paper is generally readable, but a few small issues affect clarity:
* Line 97: the summation variable t is undefined in the equation.
* The subgraph notation should be introduced before its first use (currently used at line 136 but defined at line 140).
* The expression “⇒ ρ(G′)” on line 36 is unclear — it’s not obvious what the authors intended.
* Adding a clearer description of the training and generation procedures (perhaps in pseudo-code) would also make the paper easier to follow.


[1]. Zahirnia, Kiarash, et al. "Micro and macro level graph modeling for graph variational auto-encoders." NeurIPS 2022.

[2]. Zahirnia, Kiarash, et al. "Neural graph generation from graph statistics." NeurIPS 2023.

[3]. Thompson, Rylee, et al. "On evaluation metrics for graph generative models." ICLR 2022.

[4]. Shirzad, Hamed, Kaveh Hassani, and Danica J. Sutherland. "Evaluating graph generative models with contrastively learned features." NeurIPS  2022.

**Strengths:**

Enforcing structural properties within diffusion-based graph generative models is both novel and interesting. By introducing absorbing and recurrent states into the diffusion process, the authors present a principled approach to preserving these constraints throughout generation.

**Weaknesses:**

The method involves validity checks during the reverse diffusion process, which the authors acknowledge can be computationally expensive.

Experiments are limited to small synthetic datasets; real-world datasets (e.g., molecular graphs, citation networks) are missing.

**Questions:**

How does the computational cost of enforcing constraints scale with graph size?

---

### Official Review · Reviewer_YaeU · 2025-11-09

**Soundness:** 3
**Presentation:** 2
**Contribution:** 2
**Rating:** 2
**Confidence:** 3

**Summary:**

The paper proposes a graph generation approach called ValiGraph, which extends discrete diffusion models by introducing auxiliary (buffer) states to represent “undecided” edges.
This idea allows the model to respect structural constraints such as planarity or connectivity during generation.
The authors evaluate the method on three synthetic datasets (Community, Planar, Lobster) and introduce a comparison metric based on extended persistent homology.

**Strengths:**

The focus of the paper i.e., enforcing structural constraints is relevant and well motivated.
In the literature, many graph generation models struggle to preserve properties such as planarity, which can be easily violated even by small structural changes.
The idea of explicitly modeling these constraints within the diffusion process is conceptually interesting.

**Weaknesses:**

- The experiments are conducted only on synthetic datasets with specific constraints (Community, Planar, Lobster), without any validation on real-world data such as molecular graphs, social networks, or attributed graphs.
This limits the assessment of the method’s generality and practical usefulness.

- In several cases, the performance improvements do not justify the computational cost of the proposed approach.
In practical scenarios, it might be more efficient to generate many graphs and filter out invalid ones rather than enforcing validity at every generation step.
Moreover, the quantitative results are marginally better than those of DiGress or ConStruct, and sometimes worse.

- The paper lacks any computational cost analysis and comparison with competitors

- Figure 5: it would be helpful to use different markers and/or line styles to improve readability for color-blind readers.

Typos
- “Respecting hards structural constraints” >>> “hard structural constraints”
- Figure 4 caption: “Digressexhibits” >>> “DiGress exhibits”

**Questions:**

- How is the VUN metric computed exactly?
Some of the reported values seem inconsistent, and a clarification of the calculation would be helpful.

---

### Meta-Review · Area_Chair_wEte · 2026-01-07

**Summary:**

The paper proposes ValiGraph, a discrete diffusion model designed to enforce structural constraints (such as planarity or connectivity) during graph generation. While the reviewers unanimously acknowledged the importance of the problem setting, the submission received four consistent rejection ratings (Score: 2). The primary shortcomings identified are the exclusive reliance on small synthetic datasets, the lack of analysis regarding the computational cost of validity checks, and missing comparisons to relevant recent baselines. Additionally, the clarity of the method regarding constraint enforcement mechanisms and the theoretical grounding of the proposed topological metric were significantly questioned.

**Reviewer Concerns:**

No rebuttal has been made

**Reviewer Scores:**

No rebuttal has been made

---

### Decision · Program_Chairs · 2026-01-26

Reject